# The Identification of a Yield-Related Gene Controlling Multiple Traits Using GWAS in Sorghum (*Sorghum bicolor* L.)

**DOI:** 10.3390/plants12071557

**Published:** 2023-04-04

**Authors:** Yizhong Zhang, Xinqi Fan, Du Liang, Qi Guo, Xiaojuan Zhang, Mengen Nie, Chunhong Li, Shan Meng, Xianggui Zhang, Peng Xu, Wenqi Guo, Huiyan Wang, Qingshan Liu, Yuxiang Wu

**Affiliations:** 1College of Agriculture, Shanxi Agricultural University, Taigu 030801, China; 2Sorghum Research Institute, Shanxi Key Laboratory of Sorghum Genetic and Germplasm Innovation, Shanxi Agricultural University, Yuci 030600, China; 3Shanxi Key Laboratory of Minor Crops Germplasm Innovation and Molecular Breeding, State Key Laboratory of Sustainable Dryland Agriculture (In Preparation), Shanxi Agricultural University, Taiyuan 030031, China; 4Jiangsu Academy of Agricultural Sciences, Nanjing 210014, China

**Keywords:** sorghum, six yield-related traits, GWAS, *VIP5*, breeding

## Abstract

*Sorghum bicolor* (L.) is one of the oldest crops cultivated by human beings which has been used in food and wine making. To understand the genetic diversity of sorghum breeding resources and further guide molecular-marker-assisted breeding, six yield-related traits were analyzed for 214 sorghum germplasm from all over the world, and 2,811,016 single-nucleotide polymorphisms (SNPs) markers were produced by resequencing these germplasms. After controlling Q and K, QTLs were found to be related to the traits using three algorisms. Interestingly, an important QTL was found which may affect multiple traits in this study. It was the most likely candidate gene for the gene *SORBI_3008G116500*, which was a homolog of *Arabidopsis thaliana* gene-*VIP5* found by analyzing the annotation of the gene in the LD block. The haplotype analysis showed that the *SORBI_3008G116500^hap3^* was the elite haplotype, and it only existed in Chinese germplasms. The traits were proven to be more associated with the SNPs of the *SORBI_3008G116500* promoter through gene association studies. Overall, the QTLs and the genes identified in this study would benefit molecular-assisted yield breeding in sorghum.

## 1. Introduction

*Sorghum bicolor* is one of the oldest crops cultivated by human beings, and it is also one of the only crops affected by artificial selection evolution [1]. Meanwhile, the wild sorghum that is commonly distributed in Africa is a promising resource for sorghum improvement [2]. *Sorghum* is widely distributed all over the world and has various uses, such as in food, wine making, and so on [3]. Moreover, sweet sorghum could be used for bioenergy with genetically diverse characteristics and variation exits [4]. Sweet sorghum was considered in the model system for bioenergy crops due to its favorable low-input cost traits [5]. Due to the different climate, soil setting, and cultivation system in the cultivation area, various types in sorghum have formed different agronomic, physiological, and biochemical characteristics through natural and artificial selection in the long-term cultivation process [6]. The variations of traits laid a foundation for the exploration of related genes in sorghum.

With the increase in the world population, the demand for grain production is also higher [7]. Understanding the genetic variations in traits related to yield was a necessary basis for breeding [8]. In the process of breeding, yield has always been the concern of breeders, so analyzing the traits relating to yield could provide effective guidance to breeders [9]. 

To date, the traits related to yield have been well studied in plants. In rice, plant height and the number of internodes are the key factors determining not only plant biomass, but also yield. For instance, the *GRF* (*GROWTH-REGULATED FACTOR*)*4* gene increased plant height and yield [10]; the panicle type also directly affects yield. The *DEP* (*DENSE AND ERECT PANICLE*)*1* gene of rice, which encodes a phosphatidylethanolamine-binding protein-like domain protein, could directly improve yield by affecting the panicle type [11]. Meanwhile, the heading stage can also affect the yield in rice. The *DTH* (*Days to heading*)*8* gene was isolated through the heading stage. At the same time, the *DTH8* superior haplotype could improve the yield of rice [12]. Moreover, the 100-grain weight is one of the most direct factors in yield. Many genes in plants such as *TGW* (THOUSAND-GRAIN WEIGHT)*6* could increase the yield [13]. Previous studied showed that the traits related to yield could be identified, and that the traits would be better for explaining the mechanism of yield. In sorghum, some genes which influenced the yield were cloned. The *BY*(*BIOMASS YIELD*)*1* gene encodes a 3-deoxyd-arabino-heptulosonate-7-phosphate synthase (DAHPS), which was cloned by map-based cloning and could regulate biomass and grain yield through the shikimate pathway [14]. Allelic variations in the NAC transcription factor were found to affect grain yield, sugar yield, and biomass composition in near-isogenic lines (NILs) [15]. The previous study showed that other traits, such as the leaf length, leaf breadth, number of leaves per plant, panicle weight, and hundred-seed weight may have also been given importance along with yield according to correlation analysis results [16]. Therefore, yield was a complex trait, and understanding it required inheritance variations in multiple components to be analyzed together.

Genome-wide association study (GWAS) aims to explore excellent allelic variation using natural variation populations and provide superior natural variations in materials for breeders [17,18,19,20]. Compared to bi-parental quantitative trait loci (QTL) analysis, GWAS could detect multiple alleles, save time, and offer some other advantages [8], such as the deconstruction of offspring populations. One natural variation population could be used to investigate multiple traits, and it effectively solved the problem that multiple components of yield have different genetic variations [21,22,23]. 

Meanwhile, GWASs have been widely used in sorghum. In order to improve environmental adaptation, 1943 georeferenced sorghum landraces were used for the SNP detection of experimental drought stress and aluminum toxicity [24]. Recently, researchers observed major heat tolerance traits and preformed GWASs for traits responsive to heat stress using 374 accessions [25]. Association mapping was used for germinability and seedling vigor under low-temperature conditions in sorghum. Besides the traits related to abiotic stress, the traits with seed were studied using association mapping. Li et al. used the genotyping by sequencing (GBS) method to genotype 238 mini-core collection sorghum and 7 breeding varieties, and they collected the data of forage quality-related traits. As a result, 42 SNPs were found which were associated with traits and 14 genes were predicted [26]. The grain polyphenol concentrations were studied using the global germplasm, and it was found that the flavonoid pathway loci may be useful to guide the future enhancement of cereal polyphenols [27]. To identify the natural variations in sorghum grain quality traits, 196 diverse sorghum inbred lines were used for GWASs, and 1,414,494 genetic loci were found [28]. Three traits (grain yield per primary panicle, grain number per primary panicle, and 1000-grain weight) related to yield were collected from 390 diverse accessions and were used for GWASs. As a result, the gene, namely the ethylene receptor, was predicted as the candidate gene [29]. The other trait, the nodal root angle, was analyzed using GWAS in sorghum and 58.2% of the phenotypic variance was explained [30]. In sweet sorghum, important traits, including brix and height, were conducted for association mapping [31]. The previous study above showed that GWAS could be useful to analyze the population and detect the significant SNPs in sorghum. 

Although there are some studies on the traits of yield, the genetic variation in other specific yield traits, such as plant height and the relationship between genetic variations with various traits, remains largely unknown. We herein collected a population consisting of 214 landraces from diverse countries to observe six traits related to yield. Via the resequencing method, 2,811,016 markers were called. Using the emmax, fast-lmm, and lmm algorisms with controlling Q and K, we identified numbers of SNP markers associated with the traits. Finally, the *SORBI_3008G116500* gene was found, which may control four traits through the annotation of genes in the LD block, and the elite haplotype was analyzed. 

## 2. Result

### 2.1. Population Collection and Genotyping

The landraces from different regions could provide more genetic variation; therefore, 214 landraces from China (86), India (71), America (36), Japan (14), Russia (6), and Mexico (1) were collected (Figure 1a, Appendix A). The landraces laid the foundation of genetic variation for the GWAS. After that, the population was genotyped using the resequencing method. As a result, we obtained about 2198 G raw base data, and the Q30 percentage was 92.5%. The raw reads were 14.6 Gb. The average sequencing depth was 13.4, and the average coverage ratio of 10 times of population was 71.2%. After analysis, we obtained 6,738,606 SNPs and 1,336,590 indels. The transition–transversion rate was about 1.9 in all varieties, and the average heterozygosity was 21.4%. Overall, the high-quality molecular markers derived from the germplasm were obtained based on resequencing and analyzing data, which then laid the foundation for subsequent analysis.

### 2.2. SNP Markers Selection and Population Structure Analysis

Firstly, 2,811,016 SNP markers were left after filtering with the missing rate (<0.8) and the minor allele frequency (>0.05). Through separating the markers into each chromosome, we found that the average number of SNP markers was about 281,102. The Chr.NC_012871 had the most SNPs (327,237), while chr.NC_012876 had the least SNPs (224,286) (Figure 1b, Appendix A). Based on the SNP markers, including the population structure, phylogenetic tree, PCA, and kinship analysis, admixture software was used, and the cross-validation (CV) error was calculated (Figure 1d, Appendix A). The results show that the CV error was the smallest when K = 9. The population could therefore be divided into nine subpopulations (Figure 1d). Combined with the origin of the landraces, the landrace from China showed the highest proportion in Q1, Q3, and Q5, while the landraces from India were mainly found in Q2 and Q8 (Figure 1d). Moreover, the neighbor-joining clustering of landraces based on genetic distance was analyzed, and the result was almost consistent with that of population structure (Figure 1c). Then, most varieties had distant phylogenetic relationships according to the kinship analysis (Appendix A). Lastly, PCA analysis showed almost no varieties with the same background (Appendix A, Appendix A). The above results show that the population we selected was suitable for GWASs and could be used to identify the number of variants in some loci.

### 2.3. Phenotype Analysis

In order to elucidate the genetic base of traits related to the yield of sorghums, traits including the heading date, plant height, spike type (Appendix A), spike length, number of internodes, and 100-grain weight were observed (Figure 2a–f). As a result, extensive phenotypic variations were observed in all traits. The variation coefficients in the six traits were 0.21, 0.40, 0.53, 0.28, 0.30, and 0.22. The phenotype values of plant height were 0.69 to 3.96. The spike type ranged from 1 to 5, while the spike length varied from 13.67 to 69.17. Lastly, the number of sections ranged from 2 to 22 (Appendix A, Table 1). After calculating the correlation of each trait, the number of internodes had the most relevance with the heading date, while the smallest relevance was found with the spike length (Figure 2g). The results show that the total observed six traits were abundant, but the intrinsic mechanism relating to yield formation may be different.

### 2.4. LD Decay Analysis and GWAS

To estimate the linkage disequilibrium (LD) of the population, the r^2^ was calculated using the Plink software. The variation in LD decay ranged from 9.9 to 70.2 kb, and the average was 17.4 K. The greatest LD decay in all chromosomes was found in NC_012875, while the least was found in NC_012870 (Appendix A). 

Based the collection of phenotype data and the population analysis, three GWAS algorisms were used to detect the QTLs. As results, 29 SNPs which exceeded the threshold (−log_10_(*p*) = 7.75) were observed in all six traits using the emmax (Figure 3, Appendix A). The greatest number of SNPs (21) was detected in spike length trait, while the smallest numbers of SNPs (0) were found in plant height, the number of internodes, and 100-grain weight trait. Of the 21 SNPs significantly associated with spike length, 3 were found on chromosome NC_012871.2, 1 was found on chromosome NC_012872.2, 7 were found on chromosome NC_012874.2, 6 were found on chromosome NC_012876.2, 1 was found on chromosome NC_012878.2, and 3 were found on chromosome NC_012879.2. The other three and five significant SNPs on chromosome NC_012875.2 were found in the heading date trait and on chromosome NC_012874.2 in the spike-type trait, respectively (Appendix A). With the calculation of the fastlmm model, in total, 1175 SNPs were found (Appendix A, Appendix A). The same positions were detected in the heading date and the spike-type trait. Among all the significant SNPs, the NC_012874.2 chromosome had the highest spike length (1079) (Appendix A). The SNP (NC_012874.2 1992075) was found in the plant height trait, while no SNP was found using emmax. Moreover, the lmm model of GEMMA was used to identify the 1320 associated SNPs (Appendix A, Appendix A). Luckily, there were two significant SNPs in chromosomes NC_012876.2 and NC_012878.2 (Appendix A), while no SNPs were found in the other two algorisms (Appendix A). 

The change in heading date may lead to increases in the yield [32]. By using GWAS to detect the loci controlling the heading stage, three SNPs exceeded the threshold (−log_10_(*p*) = 7.75), representing the same QTL (Figure 3). The other loci did not exceed the threshold, and it may contain genes controlling heading date. 

In the plant height trait, there was one common significant SNP (NC_012874.2 1992075) within one QTL using the fast-lmm and lmm algorisms. From the Manhattan plot, this peak was more continuous than other peaks (Figure 3). Therefore, a gene which controls the plant height would need to be cloned in future studies.

The shape and length of spike were also directly related to the yield of plants. In the GWAS results, 5 and 1307 SNPs were found to be significantly associated with spike traits using the lmm model. In panicle length traits, multiple SNPs on the NC_012874.2 chromosome were found to exceed the threshold (Figure 3). This indicates that there may be multiple genes related to ear length on this chromosome.

After detecting the significant SNPs in the two traits, no SNPs associated with the number of internodes and 100-grain weight were identified unfortunately. From the Q-Q plot, it is possible that the model did not match the two phenotypes (Appendix A). Although there were no significant SNPs, there were some peaks which showed the trend of further research, such as the QTL located in the end of NC_012878 (Figure 3). 

### 2.5. Prediction and Analysis of Yield-Related Gene

Although there were numbers of QTLs in the GWAS, it was important to find the QTLs which were identified in several traits. Additionally, we could clone genes that affected multiple phenotypes and explored excellent alleles. Luckily, there were signals at the adjacent position on the NC_012877 chromosome in the traits of heading date, plant height, spike type, and spike length (Figure 4a up). There was probably a gene with multiple functions at the locus, such as the *DTH8* gene, in rice [12]. 

To identify whether some genes were the causal genes in the QTL, we calculated the LD block. The significant SNP was in the LD block from 52,637,266 to 52,792,992 and from 52,793,087 to 53,095,551 (Figure 4a bottom). In the LD blocks, there were 23 predicted genes (Figure 4a middle, Appendix A). In all genes, the *SORBI_3008G116500* was annotated as *VIP5*. Luckily, the homologue gene of *SORBI_3008G116500* in *Arabidopsis thaliana* was reported to affect the heading date [33,34]. So, the *SORBI_3008G116500* gene was mostly the candidate gene. 

To observe the elite haplotype, three haplotypes were found according to the SNP in the promoter and gene (Appendix A). The haplotype analysis showed that hap3 was superior compared to the other two based on the four traits’ data (Figure 4b). In all varieties, Hap3 provided the least percentages compared with the other two haplotypes, suggesting that the elite haplotype was not fully used (Figure 4c). Combined with the origin of the three haplotypes, the Hap1 almost spread all over the regions where we collected the germplasm. The Hap2 was most present in China, while the elite haplotype-Hap3 only existed in China (Figure 4c). This indicates that higher polymorphism in the candidate gene was found in China. Through the gene association study, the SNP in the promoter may play a more significant role in the gene function (Figure 4d). Above that, we believed that *SORBI_3008G116500* was most likely the candidate gene in the QTL and the superior haplotype could be widely used for sorghum breeding, especially in other countries.

## 3. Discussion

### 3.1. GWAS Results and Observed Traits

In our study, the most significant SNPs were detected in the spike length trait using three algorisms. Meanwhile, the panicle length has been investigated by linkage mapping [35]. Compared with the previous QTLs, more novel QTLs were found in chromosome NC_012874.2 and so on (Appendix A). At the same time, GWAS was also carried out on the grains per panicle and grain size in sorghum [36]. In our study, many QTLs were detected using GWAS, which improved the knowledge on the genetic variations in yield, but more QTLs would be found if we observed more traits related to yield such as grain size [37] or traits related to leaf [16]. Thus, more characteristics could be added for the GWAS, and we will obtain more about the genetic variation in the configuration factors of sorghum yield, which is more conducive to the analysis of the genetic variation in each character and the aggregation analysis of the favorable allele variation. 

### 3.2. A More Appropriate Model or Lower Threshold for Detection

Association analysis is an important method used to analyze the genetic mechanism of complex traits [21]. Population structure analysis could reduce the impact of population classification on the division of subgroups in association analysis [38]. We found no significant loci in the number of internodes and 100-grain weight traits. The positive sites may be screened out due to the problem of model selection [17]. Additionally, the 100-grain weight trait may be a complex trait. In the further study, a higher *p*-value was selected for site screening. Meanwhile, 100-grain weight trait could be divided into some other related traits, such as the grain length and width, and the significant loci may be detected. Combined with gene verification, we could also clone the genes that control the traits. Finally, some other algorisms, such as restrictive two-stage multipoint (RTM) GWASs, could be used for significant SNP identification in the 100-grain weight trait [39]. Meanwhile, more careful verification was required because of the lower threshold of RTM-GWAS. In our study, the novel QTL with higher reliability was the purpose and the RTM-GWAS could be used for further study if more QTL want to be detected.

### 3.3. SORBI_3008G116500 Breeding

In sorghum, researchers have identified many QTLs with multiple traits [26,28,36,40,41], and so did we. After that, the genes were cloned by GWAS, and superior alleles and markers linked to the allele were found [42]. These genes would be beneficial for molecular breeding, such as marker-assisted selection. The *AtVIP5* functioned in the heading date trait [43]. Loss of the mutants, such as the other VIP genes (for example, *VIP2/ELF7*, *VIP4*, *VIP5*, and *VIP6/ELF8*) also flower earlier because of silencing of *FLOWERING LOCUS C* (*FLC*) [33,44,45]. In rice, the *IPI1* gene which was annotated as VIP2 modulates IPA1 protein levels, and the plant architecture, including plant height and branch number, were changed when overexpressing the *IPI1* [43]. In our study, we supposed that the homologue of *AtVIP5* could function not only in the heading date, but also in the spike trait. In the later stage, combined with transgenic lines or mutant lines, further validation would need to be conducted to understand the function of *SORBI_3008G116500*, and excellent allele markers would need to be developed for molecular marker selection in sorghum breeding, especially improving the traits including heading date, plant height, spike type, and spike length. Additionally, the Chinese sorghums have formed a rich genetic base and unique characteristics have evolved as reported previously [46]. The elite haplotype of *SORBI_3008G116500* was mostly distributed in China, validating that the sorghum germplasm in China had more potential for breeding compared with the germplasm in other countries.

## 4. Material and Method

### 4.1. Plant Materials and Environment Conditions

A total of 214 accessions comprised the varieties were used in our study. Materials No.1-No.186 were provided by Sorghum Germplasm Resource Conservation Bank of Sorghum Research Institute of Shanxi Agricultural University. Materials No.187–No.214 were collected from “The Third National Survey and Collection of Crop Germplasm Resources” Special Project for the Germplasm Resources Protection of the Ministry of Agriculture and Rural Affairs of the People’s Republic of China and were provided by the Jiangsu Academy of Agricultural Sciences. These were planted on the Liuhe base of the Jiangsu Academy of Agricultural Sciences in 2018. The test site is located at 118.63 °E and 32.48 °N. This area has a subtropical monsoon climate, with an average temperature of 15.4 °C. The average precipitation is 1073.8 mm, the average frost-free period is 225 days, and the average annual sunshine total is 1956.2 h. The standard agricultural practice was followed for plant growth and development, including irrigation, fertilizer application, and pest control. All experiments used a completely randomized block design with three replicates. The authors declare that all that permissions or licenses were obtained to collect the sorghum plant, and that all aspects of the study comply with relevant institutional, national, and international guidelines and legislation for plant ethics in the “Methods” section.

### 4.2. Genotyping

All accessions of DNA were extracted using the DNAsecure plant kit (Qiagen, Cat. No. DP320). The truseq library construction kit (FC-121-4001) was used to build the library. DNA fragments were prepared by end repair, and PloyA tail and sequencing connector were added, followed by purification and PCR amplification. The libraries were analyzed by the Bioanalyzer (Agilent, Santa Clara, CA, USA), and the PCR products were quantified by a Qubit 3.0 fluorometer (Invitrogen, Carlsbad, CA, USA). The constructed libraries were sequenced at Illumina Hiseq Xten platform to produce 150 bp paired-end reads. The sequencing reads were extracted from the raw data; reads with an adaptor, low-quality reads, and reads with an unidentified nucleotide sequence were removed using fastp-0.20.1 (http://opengene.org/fastp/fastp, accessed on 3 December 2021) [47]. After that, the BWA and GATK software were used for alignment and variant calling for high-quality reads [48,49]. The reference was downloaded from NCBI website (ftp.ncbi.nlm.nih.gov/genomes/all/GCF/000/003/195/GCF_000003195.3_Sorghum_bicolor_NCBIv3, accessed on 3 December 2021). Moreover, we used Plink to filter the variants [50]. On the premise of minimum allele frequency (MAF) ≥ 0.05 and missing data ≤ 0.2, 2,811,016 high-quality SNPs were obtained. 

### 4.3. Phenotypic Trait Evaluation and Data Analysis

The heading date was collected at the heading stage. At the maturity stage, the other phenotypic data of five yield-related traits were collected. For collecting plant height trait, five plants were randomly selected to be measured. Numbers of Internode were determined after stripping leaves from the stem. The data of spike type was collected according to the “Guidelines for the conduct of tests for distinctness, uniformity and stability-Sorghum (*Sorghum bicolor* (L.) Moench)” in China. However, during specific measurements, the sorghum spike phenotype has a phenotype that resides between two types. We have defined the specific type in the Appendix A. According to the guidelines for conducting tests for distinctness, uniformity, and stability-Sorghum (*Sorghum bicolor* (L.) Moench) [GB/T 19557.15-2018], spike length was measured from the leaf scars under the spike to the top of the main spike. Lastly, mature grains of each inbred line were harvested and dried at 50 °C until the weight became constant. After drying, we collected the 100-grain weight data. The means of all phenotypic data and correlation coefficients were calculated using SAS.

### 4.4. Population Structure, Relative Kinship, Principal Component Analysis (PCA), and Linkage Disequilibrium (LD)

Population structure (Q) was determined using Admixture software and filtered SNP markers [51]. The number of subpopulations was determined by the CV error. Relative pairwise kinship (K) was calculated using TASSEL5.0 [52]. The PCA was analyzed with SNP markers using the TASSEL 5.0, and the results were visualized using R (https://www.r-project.org, accessed on 25 May 2022). The LD was also calculated with Plink using the SNP data of whole genomes and each chromosome. 

### 4.5. Genome-Wide Association Study (GWAS) and QTL analysis

The GWAS for the six yield-related traits was performed for the three algorisms, including lmm, fastlmm and emmax, using GEMMA, FaST-LMM, and EMMAX programs [53,54,55]. The significant value was decided by 0.05/markers. The results were visualized using R package-CMplot. The LD block was defined with LDBlockShow software [56]. The gene annotation was based on Gramene (https://www.gramene.org/, accessed on 29 July 2022). The gene association study was conducted using TASSEL5.0 [57].

## 5. Conclusions

Overall, six traits related to yield were analyzed by three different algorisms based on natural variations in 214 landraces, and many significant loci were obtained. We also analyzed the differences of the three algorisms and the detection ability for different traits. Finally, a candidate gene was obtained that could affect multiple traits and find the elite haplotype. Our results enrich the discovery of sorghum-associated SNPs, and the discovered QTLs and gene will be helpful to sorghum breeding.

## Figures and Tables

**Figure 1 plants-12-01557-f001:**
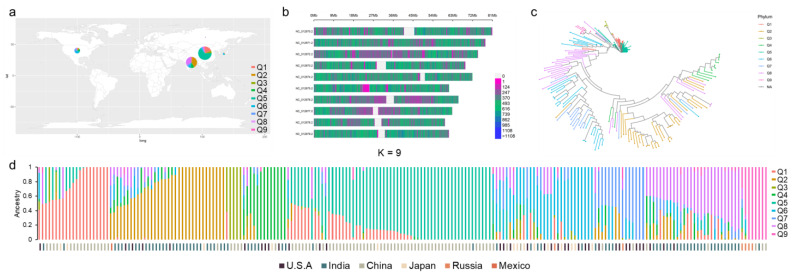
The analysis of population. (**a**) The distribution of all 214 landraces. (**b**) The density of SNPs in all chromosomes. (**c**) The neighbor−joining tree of all landraces. The colored subsections within each vertical bar indicate membership coefficient (Q) of the accession to different clusters. (**d**) The populations were pre−determined with K = 9 based on ADMIXTURE analysis. The bottom was the country of origin for each landrace. Different color indicates different membership coefficient (Q) in (**a**,**c**,**d**).

**Figure 2 plants-12-01557-f002:**
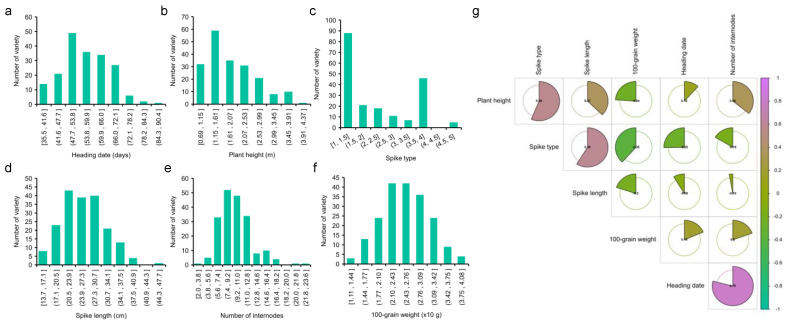
The analysis of phenotype. (**a**–**f**) The frequency distribution histogram of six traits. (**g**) The relevance of each trait.

**Figure 3 plants-12-01557-f003:**
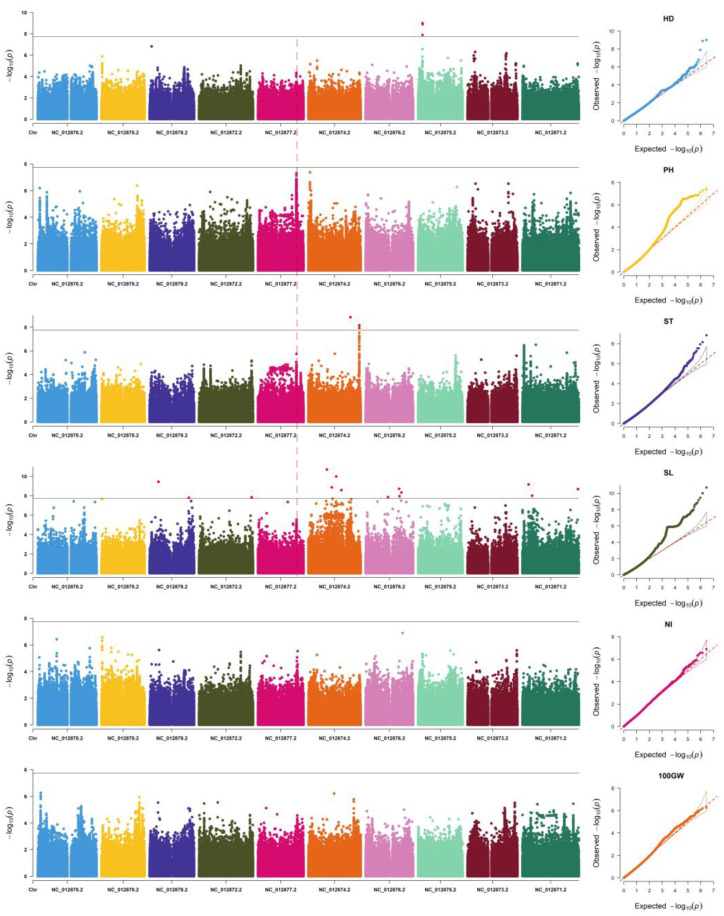
The Manhattan and Q-Q plot of six traits, including heading date (HD), plant height (PL), spike type (ST), spike length (SL), number of internodes (NI), and 100-grain weight (100 GW), based on the emmax algorism. The light red dash line indicates the QTL that was identified in four traits.

**Figure 4 plants-12-01557-f004:**
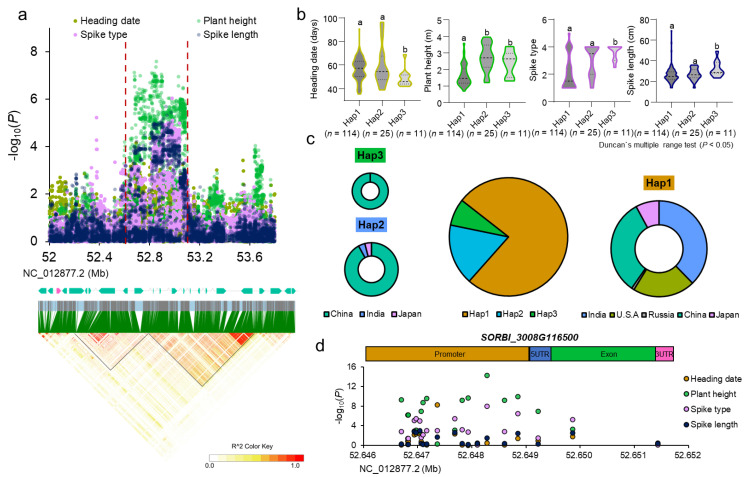
The analysis of the QTL. (**a**) Local Manhattan plot (top), the gene in the LD block (middle), and LD heatmap (bottom) surrounding the peak on chromosome NC_012877.2. The red dashed line indicates the candidate region for the peak. (**b**) The haplotype analysis of the *SORBI_3008G116500* gene. Different letters represent significant differences (P < 0.05) using Duncan`s multiple range test. (**c**) The geographical distribution of different haplotypes. (**d**) The gene association study of *SORBI*_−_*3008G116500*.

**Table 1 plants-12-01557-t001:** The minimum (Min), maximum (Max), mean values (Mean), and coefficient of variation (CV) of heading date (days), plant height (m), spike type, spike length (cm), number of internodes, and 100-grain weight (g) in the population.

Trait	Min	Max	Mean	CV
Heading date (days)	35.50	104.90	57.69	0.21
Plant height (m)	0.69	3.96	1.90	0.40
Spike type	1.00	5.00	2.34	0.53
Spike length (cm)	13.67	69.17	26.75	0.28
Number of internodes	2.00	22.00	9.79	0.30
100-grain weight (×10 g)	1.11	4.03	2.56	0.22

## Data Availability

The resequencing data were uploaded in the NCBI database (PRJNA826146, https://www.ncbi.nlm.nih.gov/sra/?term=PRJNA826146, accessed on 30 November 2022). The datasets supporting the conclusions of this article are included within the article and its additional files.

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
