# Peer review of "The Identification of a Yield-Related Gene Controlling Multiple Traits Using GWAS in Sorghum (Sorghum bicolor L.)"

_plants, 2023, doi:10.3390/plants12071557_

Round 1

Reviewer 1 Report

Identification of a yield-related gene controlling multiple traits using GWAS in sorghum (Sorghum bicolor L.) by Zhang et al.

In this manuscript, the tried to elucidate genetic basis of some traits of sorghum by GWAS approach. However, I personally feel this study lacks a clear scientific target except SORBI_3008G116500 gene. The function of this gene is also not well-characterized. Results are poorly explained preventing the understanding of general readers. I think the manuscript needs to be substantial rewrite to make it more explanatory about the results obtained by the study. Let me humbly to say that the authors can try more one-by-one explanation and discussion of their results to facilitate the understanding of the readers about the novel facts (tor comparison of the facts) obtained by this study.

Major comments:

L143-: This paragraph must describe the complete method for collecting traits.

L211-213, “The result showed that the total observed six traits could all influence the yield, but the intrinsic mechanism relating with yield formation may be different.”: Why the authors could say this without actual yield data? In addition, “intrinsic” was a fact before this study. 

Figure 2: The frequency distributions of a, c and d doesn’t seem normal, and this kind of distribution might have caused a problem in GWAS or QTL detection. The authors could consider collecting this by some kinds of function or removing outliers.

Figure 3 (and S6, S6): The beautiful circular Manhattan plots doesn’t help the readers understanding at all, because they completely lost details.

L299-307: This “discussion” is not based on this study.

Minor comments:

All the figures don’t seem to have sufficient resolution.

L31, (L.): Are this parenthesis necessary? No () in the title.

L38-41, “Due to …”: For what does this sentence mean?

L42-46: This paragraph can be merged with the previous paragraph.

L47-67: For what does this paragraph stand? Information on rice is not very helpful for this study, because the authors did not perform synteny analysis. Most of the readers expect known genes of sorghum here.

L68-, “Genome wide association study …”: I think they cannot be mixed because GWAS is different from LD analysis in their philosophy and mathematical models.

L68-: Again, too much for rice. 

L107, “emmax, fast-lmm and lmm…”: Please clarify what they specify. I mean the authors are mixedly using the names of algorisms, software packages, or libraries. 

L113-: Please specify the serial numbers (No. 1 to 214?) in the list of the materials. The material list can contain providing institutions. I mean a more comprehensive list of the materials is needed.

L129-: Specify the reference sequence and the gene models.

L131, “truseq”: Is it an illimina’s kit? Then, please specify. There are several kinds of TruSeq kit, especially the kit is PCR-free or not.

L158- “The GWAS for … lmm, fast-lmm and emmax using the GEMMA, FaST-LMM, and EMMAX software.”: Is this explanation appropriate? References are needed.

L162- “The gene association”: For what does this mean?

L170, “As results, we got about 2,198 G raw base data and the Q30 percentages was 170 92.5%. The raw reads were 14.6 Gb”: For what does this mean?

L195, “distant”: This word had a bigger font size.

L195-198, “Then, …”: For what do these sentences mean? It is not understandable as a description of the results.

Figure 1. The figure legend is not sufficient to help readers. Colors or legends (Q1-Q9) are not clearly explained.

L201, “spike type”: What is this trait? Is it quantitative?

L204-209, “The variance…”: Same information as Table 1, and variance is not easy for readers to take here.

L266, “Yield-related Gene Prediction and Analysis”: This sounds strange as an English.

L309, “GWAS Results and Observed Traits for Yield”: I can’t agree the authors really did a genetic analysis for yield.

L310-312, “Meanfile, …”: This sentence doesn’t look a discussion.

L313-324: These descriptions are not based on the data obtained by this study. I mean, the authors say just they found many QTLs. 

L326-331: The authors could discuss about the heritability of the traits without QTL. Because the authors had replication, it is possible to calculate broad sense (and it is almost the same as narrow sense) heritability.

L335: If the authors knew the RTM-GWAS, why they didn’t do?

L337-346: This section doesn’t contain much scientific information obtained by this study though the references help. This section should be directed to the function (not only molecular function but also phenotypic). 

Reviewer 2 Report

The study investigated 214 sorghum germplasm and identified several QTLs related to yield traits by GWAS. Then they found a candidate gene, which is the homolog of VIP5 in Arabidopsis. The study is comprehensive and the results are clear. The figures are pretty nice and well-organized. I have some suggestions for improvements.

1.       English writing could be improved. There are some typos in the manuscript and please revise them carefully. For example, in line 192, “Q1, Q3mand Q5.”

2.       Fig. 1a. What are the color differences in the pi-graph? Please indicate them clearly in the legend.

3.       Table 1. I do not understand how the spike type is measured. And also in table 1, the max for spike type is 5, but in line 207, the spike type was from 1 to 4. Please add some description for the measurement for the spike type.

4.       Fig 4. At the bottom of the figure, what does NC_012877.2 (mb) belong to? 4a or 4d?

Round 2

Reviewer 1 Report

Identification of a yield-related gene controlling multiple traits using GWAS in sorghum (Sorghum bicolor L.) by Zhang et al.

Unfortunately, it seems the authors didn’t consider the previous review very seriously. I can’t find meaningful improvements on some of my previous review. Let me repeat the same comments as previous, most of major and 2 of the minor comments.

Major comments:

V1: L143-: This paragraph must describe the complete method for collecting traits.

V2: 2.3: “The data of spike type was collected ac- cording to “Guidelines for the conduct of tests for distinctness, uniformity and stability- Sorghum (Sorghum bicolor (L.) Moench)” in China.”: This doesn’t provide sufficient information. Even whether it is quantitative is not clear because the trait is a kind of “type”. Does the “spike type” mean endosperm type? Even if so, a comprehensive explanation is needed for general readers.

V2: 2.3: “Spike length was measured from the leaf scars at the end of the spike to the top of the spike.” I am sorry I can’t understand this. What does “spike” mean, spikelet or panicle?

L211-213, “The result showed that the total observed six traits could all influence the yield, but the intrinsic mechanism relating with yield formation may be different.”: Why the authors could say this without actual yield data? In addition, “intrinsic” was a fact before this study. 

V2: 3.3: “The results show the total observed six traits had abundant, but different variations, but the intrinsic mechanism relating to yield formation may be different.”: I can’t understand “different variation”, for what does this mean?

Figure 2: The frequency distributions of a, c and d doesn’t seem normal, and this kind of distribution might have caused a problem in GWAS or QTL detection. The authors could consider collecting this by some kinds of function or removing outliers.

No response from the author.

Figure 3 (and S6, S6): The beautiful circular Manhattan plots doesn’t help the readers understanding at all, because they completely lost details.

It seems the dotted red line hides the peaks.

L299-307: This “discussion” is not based on this study.

V2: L309-316, 4.0?: The author don’t provide any scientific insights about correlations.

Minor comments:

L335: If the authors knew the RTM-GWAS, why they didn’t do?

It seems there is no change.

L337-346: This section doesn’t contain much scientific information obtained by this study though the references help. This section should be directed to the function (not only molecular function but also phenotypic). 

V2: 4.3: It seems the authors failed to provide sufficient information/evidence why this gene has such pleiotropy. I think a deeper insight could be obtained.
